# An intracellular anion channel critical for pigmentation

**Nicholas W Bellono[1], Iliana E Escobar[1], Ariel J Lefkovith[2,3], Michael S Marks[2,3], Elena Oancea[1]\***

[1]Department of Molecular Pharmacology, Physiology and Biotechnology, Brown University, Providence, United States; [2]Department of Pathology and Laboratory Medicine, Children's Hospital of Philadelphia, University of Pennsylvania, Philadelphia, United States; [3]Department of Physiology, Children's Hospital of Philadelphia, University of Pennsylvania, Philadelphia, United States

**Abstract** Intracellular ion channels are essential regulators of organellar and cellular function, yet the molecular identity and physiological role of many of these channels remains elusive. In particular, no ion channel has been characterized in melanosomes, organelles that produce and store the major mammalian pigment melanin. Defects in melanosome function cause albinism, characterized by vision and pigmentation deficits, impaired retinal development, and increased susceptibility to skin and eye cancers. The most common form of albinism is caused by mutations in oculocutaneous albinism II (OCA2), a melanosome-specific transmembrane protein with unknown function. Here we used direct patch-clamp of skin and eye melanosomes to identify a novel chloride-selective anion conductance mediated by OCA2 and required for melanin production. Expression of OCA2 increases organelle pH, suggesting that the chloride channel might regulate melanin synthesis by modulating melanosome pH. Thus, a melanosomal anion channel that requires OCA2 is essential for skin and eye pigmentation.

**\*For correspondence:** elena_oancea@brown.edu

**Competing interests:** The authors declare that no competing interests exist.

## Introduction

Ion channels are membrane proteins that regulate the concentration of key signaling ions to control a wide range of cellular functions. While plasma membrane ion channels have been extensively studied, much less is known about the identity and physiology of intracellular channels because they are less accessible to direct electrophysiological characterization.

Melanosomes are lysosome-related organelles that produce and store melanin, a natural pigment present in most organisms. Impaired melanin synthesis and storage affects visual system development and pigmentation of the skin, eyes, and hair, leading to reduced protection against ultraviolet radiation and predisposition for skin and eye cancers. A number of genes encoding putative melanosomal ion transport proteins are critical for melanosomal function, as mutations in these genes result in oculocutaneous albinism (OCA) (*Montoliu et al., 2014*). This suggests that ionic homeostasis plays an important role in melanin synthesis and storage, yet how ion channels might contribute to melanosome function and pigmentation remains poorly understood (*Bellono and Oancea, 2014*).

One of the most common forms of albinism is caused by mutations in a highly conserved protein encoded by the oculocutaneous albinism II gene (OCA2) (*Gardner et al., 1992*; *Rinchik et al., 1993*; *Rosemblat et al., 1994*; *Lee et al., 1994a*; *Sitaram et al., 2009*) (*Figure 1—figure supplement 1*). OCA2-deficient animals lack pigment (*Gardner et al., 1992*; *Protas et al., 2006*) and reduced OCA2 expression leads to decreased melanin underlying blue eye color in humans (*Eiberg et al., 2008*; *Sturm et al., 2008*). OCA2 has twelve predicted transmembrane domains (*Gardner et al., 1992*), is localized to melanosomal membranes (*Rosemblat et al., 1994*; *Sitaram et al., 2009*), and has been

**eLife digest** Melanin is a pigment found in our skin, eyes and hair. Individuals who are unable to make or store melanin, a condition known as albinism, have unusually pale features and problems with vision. The pigment helps to protect us from harmful UV radiation, and so individuals with albinism also have an increased risk of developing skin and eye cancers.

In cells, melanin is made and stored in compartments called melanosomes. The most common type of albinism is caused by defects in a protein called OCA2, which is found in the membrane that surrounds melanosomes. However the role of OCA2 in melanin production is unclear.

It has been proposed that OCA2 may allow charged particles (or ions) to enter or leave melanosomes. Here, Bellono et al. used a technique called patch-clamp to study the movement of ions across the membrane of melanosomes from skin and eye cells. The experiments show that a flow of chloride ions out of the melanosome is required for melanin to be produced. OCA2 is involved in the ion movement, and it might alter the acidity of the melanosome when present.

Bellono et al. propose that OCA2 is part of an ion channel that allows chloride ions to pass through the membrane, to make the melanosome less acidic and enable melanin to be produced. The next challenge will be to identify other ion channels in the melanosome and understand their roles in producing melanin.

implicated in pH regulation of melanosomes and trafficking of the melanogenic enzyme tyrosinase (*Puri et al., 2000*; *Manga and Orlow, 2001*; *Chen et al., 2002, 2004*; *Ni-Komatsu and Orlow, 2006*). Despite its importance, the function of OCA2 and the molecular mechanism by which it regulates melanin are not known.

Here we identify and characterize a new intracellular ion channel that resides in the melanosomal membrane and requires OCA2 expression. Using whole-organelle and single-channel patch-clamp recordings we found that OCA2 contributes to an anion channel required for pigmentation. The OCA2-mediated $Cl^-$ current was nearly abolished by a mutation identified in patients with oculocutaneous albinism type II. Interestingly, expression of OCA2 in endolysosomes increased organelle pH, providing a potential mechanism for how OCA2 regulates melanogenesis. Thus, a previously uncharacterized OCA2-dependent anion channel is critical for melanosomal function and pigmentation, revealing a novel function for intracellular ion channels.

## Results and discussion

Because intracellular ion channels are important regulators of organellar and cellular function and defects in melanosomes, lysosomal-related organelles present in melanocytes and retina, often result in severe pigmentation phenotypes, we wondered how ion channels contribute to melanosome function. We investigated OCA2, a melanosome-specific membrane protein of unknown function that, when mutated, results in albinism.

### OCA2 expression in endolysosomes leads to a $Cl^-$ conductance ($I_{OCA2}$)

In melanocytes and retinal pigment epithelium (RPE) OCA2 is restricted to melanosomes, but when expressed in heterologous systems it localizes to lysosomes and late endosomes (endolysosomes) (*Sitaram et al., 2009*). To investigate if OCA2 has ion transport activity, we expressed OCA2 tagged with GFP (GFP-OCA2) or mCherry (mCherry-OCA2) in AD293 cells, where it colocalized with the endolysosomal marker LAMP1 (*Figure 1A*). Endolysosomes are an ideal melanosome-related heterologous system because they can be enlarged by treating cells with 1 µM vacuolin-1 and mechanically released from the cytoplasm with a glass pipette (*Cerny et al., 2004*; *Saito et al., 2007*; *Dong et al., 2008*). We recorded whole-organelle currents from endolysosomes expressing GFP-OCA2 and found that voltage pulses evoked a large outwardly rectifying current ($I_{OCA2}$) that was not present in mock-transfected cells (*Figure 1A*).

The measured outward current could result either from $K^+$, the main cation in the bath, moving into the endolysosome, or $Cl^-$, the main anion in the pipette solution, moving out of the endolysosome. We tested if $I_{OCA2}$ carries $K^+$ into the lumen by replacing it with $Na^+$ in the bath solution. We detected no significant change in the current amplitude or reversal potential ($E_{rev}$) (*Figure 1—figure supplement 2A,B*),

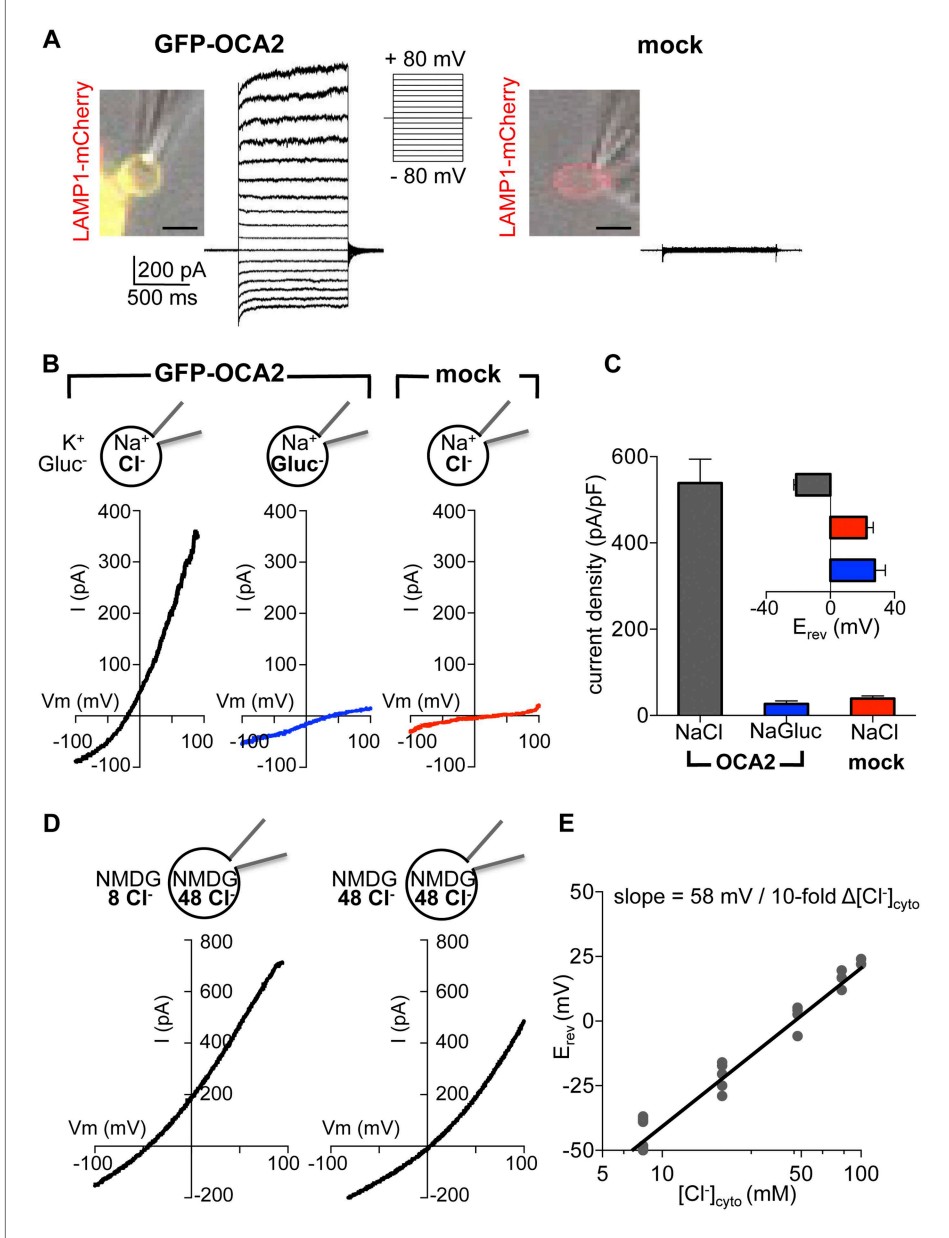

**Figure 1**. OCA2 contributes to an endolysosomal chloride current with ion channel properties. (**A**) Heterologous GFP-OCA2 localized to LAMP1-mCherry-positive late endosomes and lysosomes (endolysosomes) individually dissected from AD293 cells for patch-clamp experiments (scale bar = 5 µm). Whole-endolysosomal currents elicited by voltage steps between −80 mV and +80 mV in a representative endolysosome expressing GFP-OCA2 or in an endolysosome from a mock-transfected cell. (**B**) Representative whole-endolysosome current–voltage (I–V) relationships in response to voltage ramps. Outwardly rectifying $I_{OCA2}$ was nearly abolished and its reversal potential ($E_{rev}$) shifted in the positive direction when luminal $Cl^-$ was substituted for gluconate ($Gluc^-$), similar to currents measured from mock-transfected endolysosomes containing a $Cl^-$-based luminal solution. (**C**) Average current densities (pA/pF) measured at 100 mV. *Inset:* Average $E_{rev}$ (±s.e.m., n = 6–9 endolysosomes for each condition). (**D**) In a representative OCA2-expressing endolysosome, $E_{rev}$ was dependent on cytoplasmic $[Cl^-]$ ($[Cl^-]_{cyto}$). NMDG-based solutions were used to inhibit endogenous cation permeabilities. (**E**) $I_{OCA2}$ $E_{rev}$ varied linearly with $[Cl^-]_{cyto}$ when $[Cl^-]_{luminal}$ was kept at 48 mM. Each point represents one endolysosome at the $[Cl^-]_{cyto}$ indicated on the x-axis.

The following figure supplements are available for figure 1:

**Figure supplement 1**. OCA2 is a highly conserved transmembrane protein expressed in pigment cells.

*Figure 1. Continued on next page*

*Figure 1. Continued*

**Figure supplement 2**. $I_{OCA2}$ is not mediated by $K^+$ flux or regulated by pH.

**Figure supplement 3**. $I_{OCA2}$ is mediated by $Cl^-$ flux.

**Figure supplement 4**. $Cl^-$ channel blockers DIDS, NFA, and NPPB do not affect $I_{OCA2}$.

---

suggesting that $K^+$ is not required for $I_{OCA2}$. We next tested if $I_{OCA2}$ is mediated by luminal $Cl^-$ moving into the cytoplasm by replacing $Cl^-$ in the pipette solution with the impermeant anion gluconate (Gluc$^-$). $I_{OCA2}$ was nearly abolished with Gluc$^-$ in the pipette and $E_{rev}$ was significantly shifted in the positive direction, consistent with the endogenous current recorded in endolysosomes from mock-transfected cells (*Figure 1B,C*). The similar current amplitude and positive $E_{rev}$ measured in mock-transfected and OCA2-expressing endolysosomes with luminal Gluc$^-$ suggests the presence of an endogenous cationic permeability. When we subtracted this endogenous current from $I_{OCA2}$, the $E_{rev}$ for $I_{OCA2}$ became similar to $E_{Cl}$ predicted by the Nernst equation (*Figure 1—figure supplement 3*). The significant reduction in current amplitude in the presence of luminal Gluc$^-$ indicates that $I_{OCA2}$ is mediated by $Cl^-$, consistent with OCA2 homology to bacterial anion transporters (*Rinchik et al., 1993*; *Brilliant, 2001*; *Kobayashi et al., 2006*). We tested whether a range of pharmacological inhibitors of different $Cl^-$ channels and transporters affect $I_{OCA2}$ by bath-applying 4,4'-diisothiocyanato-2,2'-stilbenedisulfonic acid disodium salt (DIDS), niflumic acid (NFA), or 5-Nitro-2-(3-phenylpropylamino) benzoic acid (NPPB), but none significantly altered $I_{OCA2}$ amplitude (*Figure 1—figure supplement 4*). Thus, $I_{OCA2}$ has a different pharmacological profile from the channels and transporters that are inhibited by the tested compounds.

The Nernst equation predicts that if $I_{OCA2}$ were due to selective transport of $Cl^-$, $E_{rev}$ will be dependent on the $Cl^-$ concentration ([$Cl^-$]) gradient across the endolysosomal membrane. To establish whether this is the case, we determined $E_{rev}$ for currents measured at variable cytoplasmic and constant luminal [$Cl^-$], using NMDG-based solutions to prevent endogenous cationic permeability. $E_{rev}$ for 48 mM luminal and 8 mM cytoplasmic [$Cl^-$] was $-42.7 \pm 2.3$ mV and shifted to $1.6 \pm 2.5$ mV under symmetrical 48 mM [$Cl^-$] (*Figure 1D*). $E_{rev}$ increased linearly as a function of cytosolic [$Cl^-$] ([$Cl^-$]$_{cyto}$), with a 10-fold change in [$Cl^-$]$_{cyto}$ corresponding to a shift in $E_{rev}$ of 58 mV, consistent with a highly $Cl^-$ selective current (*Figure 1E*).

## Mutations associated with oculocutaneous albinism disorder affect $I_{OCA2}$

OCA2 shares little homology with known chloride channels or transporters; OCA2 might be an accessory subunit of a $Cl^-$ transporter or form a $Cl^-$ channel or carrier protein itself. To determine if specific OCA2 residues are required for $Cl^-$ transport, we sought to identify mutations important for ion transport, but not melanosomal localization. We analyzed mCherry-OCA2 variants containing disease-related mutations within highly conserved regions of the protein (*Figure 1—figure supplement 1A*). We chose mutations identified in patients with OCA type II through human genetic studies: V443I, a common albinism-associated mutation in a predicted luminal loop and important for melanin content in vitro (*Lee et al., 1994a*, *1994b*; *Sviderskaya et al., 1997*; *King et al., 2003*; *Garrison et al., 2004*; *Hongyi et al., 2007*; *Preising et al., 2007*; *Hutton and Spritz, 2008*; *Rimoldi et al., 2014*) and K614E, in a predicted cytoplasmic loop (*Passmore et al., 1999*). We also generated an OCA2 variant with five point mutations in the same predicted luminal loop as V443I (5mut: V443I, M446V, I473S, N476D, N489D) (*Lee et al., 1994a*, *1994b*; *Spritz et al., 1995*; *Hongyi et al., 2007*; *Preising et al., 2007*) (*Figure 2A*).

Upon expression in HeLa cells, the K614E and V443I mutants colocalized with the endolysosomal marker LAMP1, similar to WT mCherry-OCA2 (*Figure 2—figure supplement 1A*), indicating that both mutants traffic efficiently to endolysosomes when expressed in non-melanocytic cells. By contrast, 5mut OCA2 localized primarily to endoplasmic reticulum-like structures rather than endolysosomes (*Figure 2—figure supplement 1A*), therefore it could be used as a negative control. In pigment cells containing melanosomes (melan-ink4a), expression of OCA2 variants revealed that WT, as well as V443I and K614E mutants colocalized with the melanosomal tyrosinase-related protein 1 (TYRP1; overlap with TYRP1 = $60.1 \pm 21.3\%$ for WT; $50.7 \pm 13.9\%$ for V433I; and $53.6 \pm 16.1\%$ for K614E

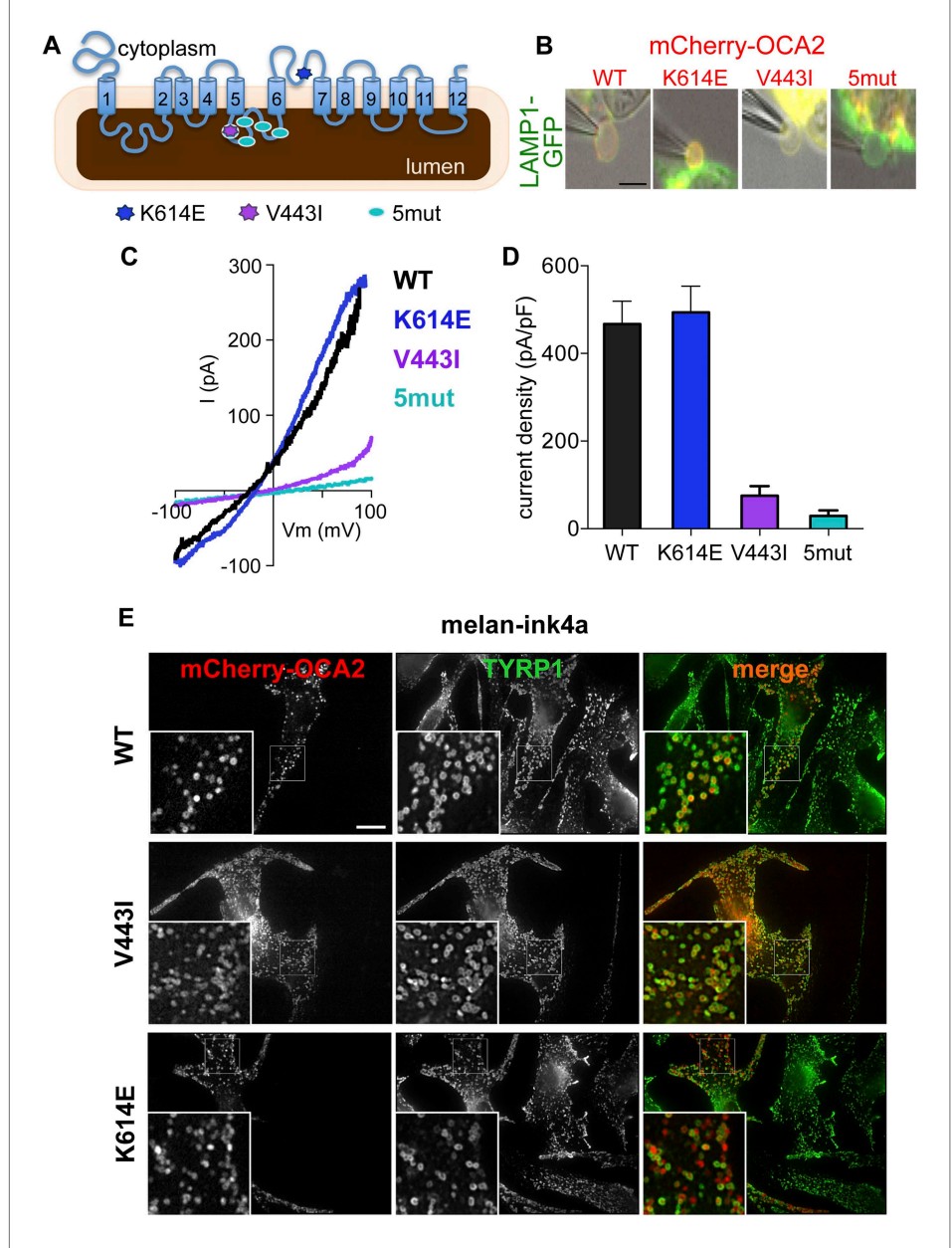

**Figure 2**. Effect of OCA2 disease-associated mutations on $I_{OCA2}$. (**A**) Predicted topology of OCA2 with albinism-associated mutations: K614E (dark blue) in a conserved cytoplasmic loop, V443I (purple) in a highly conserved luminal loop, and 5mut (light blue) consisting of 5 point mutations (V443I, M446V, I473S, N476D, N489D) in the same luminal loop. (**B**) mCherry-tagged WT, K614E, and V443I OCA2 localized to LAMP1-GFP-positive isolated endolysosomes (merged in yellow), while 5mut did not (scale bar = 5 μm). (**C**) Representative I–V relationships measured in response to voltage ramps from endolysosomes expressing WT, K614E, V443I, or from endolysosomes isolated from cells expressing 5mut OCA2. (**D**) Average $I_{OCA2}$ current densities (pA/pF) measured at 100 mV (±s.e.m., n = 4–8 endolysosomes for each condition). (**E**) Melan-ink4a melanocytes expressing mCherry-tagged (red) WT, V443I, or K614E and stained with anti-tyrosinase-related protein one (TYRP1) antibodies (green). Insets, 3× magnification of boxed regions. Merged images (right) show that WT, V443I, and K614E OCA2 variants localize primarily to TYRP1-positive compartments (scale bar = 10 μm).

The following figure supplement is available for figure 2:

**Figure supplement 1**. Localization of wild-type (WT), K614E, V443I, and 5mut OCA2.

mCherry-OCA2) (*Figure 2E*), whereas overlap with the endolysosomal marker LAMP2 was minimal (11.8 ± 10.7% among all variants). A portion of K614E localized to vesicular structures without TYRP1 but that contained pigment when imaged by bright field microscopy.

To determine if albinism-associated mutations affect $I_{OCA2}$, we expressed mCherry-tagged WT, K614E, V443I or 5mut OCA2 in AD293 cells together with LAMP1-GFP (*Figure 2B*). Currents recorded from AD293 endolysosomes identified by LAMP1-GFP and expressing K614E had similar amplitudes and $E_{rev}$ as WT OCA2 (*Figure 2C,D*), but those expressing V443I had amplitudes reduced by ~85% and $E_{rev}$ shifted to more positive values (*Figure 2C,D*). Endolysosomes of cells expressing 5mut had very small current amplitudes and a positive $E_{rev}$, similar to mock-transfected cells (*Figure 2C,D*).

Collectively, these data suggest that the V443I-containing luminal loop between transmembrane domains 5 and 6 is critical for the OCA2-mediated Cl⁻ current. The K614E mutation had little effect on localization or OCA2-mediated currents in our experiments. Because K614E was identified in albinism patients with an additional OCA2 mutation (*Passmore et al., 1999*), its associated phenotype might be too mild to detect in our assays or might be masked by overexpression.

## OCA2 regulates organelle pH

How does the OCA2-mediated anion conductance affect pigmentation? Early stage melanosomes are highly acidic (*Raposo et al., 2001*), but because tyrosinase is inactive at pH < 6.0, melanosomal pH is thought to increase in order to allow for melanin synthesis (*Ancans et al., 2001*; *Halaban et al., 2002*). We tested the hypothesis that OCA2-mediated anion extrusion from melanosomes regulates luminal pH (*Puri et al., 2000*; *Brilliant, 2001*; *Manga and Orlow, 2001*; *Chen et al., 2002*, *2004*; *Ni-Komatsu and Orlow, 2006*), similar to other anionic conductances (*Stauber and Jentsch, 2013*). We expressed in endolysosomes of AD293 cells ecliptic pHluorin-LAMP1, which becomes fluorescent at pH > 6 (*Rak et al., 2011*) (*Figure 3A*). Endolysosomes of control cells were acidic and lacked fluorescence emission at baseline (*Figure 3B*), but exhibited increased fluorescence upon neutralization (pH > 6) of organellar pH by cellular treatment with the vacuolar H⁺-ATPase inhibitor bafilomycin A1 (BafA1) (*Figure 3A–C*). In contrast, coexpression of WT OCA2 with pHluorin-LAMP1 resulted in endolysosomes that were fluorescent at baseline, with only a small increase in fluorescence in response to BafA1 (*Figure 3B,C*). This indicates that OCA2 expression increased the pH of endolysosomes to > 6.

To determine if changes in pH required the OCA2-mediated anion conductance, we examined changes in pHluorin-LAMP1 fluorescence elicited by the expression of albinism-associated OCA2 mutants. Endolysosomes expressing K614E exhibited basal fluorescence and little change in fluorescence following treatment with BafA1 (*Figure 3B,C*), consistent with its intact conductance and localization to endolysosomes (*Figure 2*, *Figure 3—figure supplement 1*). However, endolysosomes expressing V443I, which have significantly reduced current amplitudes but intact localization (*Figure 2*, *Figure 3—figure supplement 1*), had dim fluorescence at baseline that increased dramatically following BafA1 treatment (*Figure 3B,C*). Endolysosomes from cells expressing mislocalized 5mut lacked basal fluorescence and increased their fluorescence with BafA1 treatment (*Figure 3B,C*), similar to untransfected cells. Together, these results support the notion that OCA2-mediated ion transport in organelles modulates luminal pH.

We confirmed the observed OCA2-associated changes in luminal pH using the ratiometric pH-sensitive dye LysoSensor DND-160. Based on LysoSensor calibration, the luminal pH of LAMP1-positive compartments in control cells had an approximate pH value of 5.12 ± 0.03, while the luminal pH of WT OCA2-expressing endolysosomes was significantly greater (6.67 ± 0.03, *Figure 3—figure supplement 1A–C*). Expression of the K614E mutant increased pH to 6.32 ± 0.03, while V443I only modestly raised the luminal pH of endolysosomes to 5.72 ± 0.04, (*Figure 3—figure supplement 1B*). Melanosomal pH measurements were not possible because fluorescent indicator uptake in melanosomes is impaired and melanin interferes with the emission of fluorescent proteins.

Our pH measurements are consistent for the two indicators and show that OCA2-mediated Cl⁻ transport shifts the endolysosomal pH toward neutral values that in melanosomes would be optimal for tyrosinase activity and melanin synthesis. The mechanism by which OCA2 modulates pH is unclear. We hypothesize that OCA2-mediated Cl⁻ efflux from the lumen regulates the organelle membrane potential, reducing vacuolar H⁺-ATPase activity and resulting in less H⁺ being pumped in the lumen. Alternatively, the pH modulation by the OCA2-mediated Cl⁻ conductance could be a more complex mechanism involving the contribution of additional channels and transporters. Thus, our model for

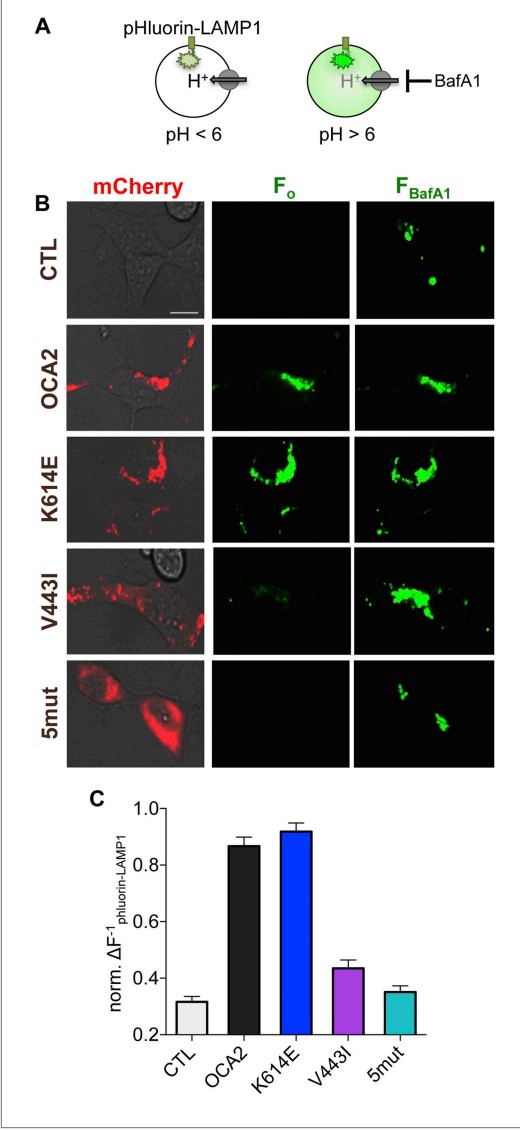

**Figure 3**. OCA2 expression regulates organellar pH. (**A**) Ecliptic pHluorin-LAMP1 fluoresces when luminal pH > 6, but not when pH < 6. The V-ATPase inhibitor bafilomycin A1 (BafA1, 2 μM) was used to neutralize acidic endolysosomes and detect pHluorin-LAMP1 expression. (**B**) Endolysosomes from mock-transfected control (CTL) AD293 cells expressing pHluorin-LAMP1 were not fluorescent at baseline ($F_o$) but brightly fluoresced after BafA1 treatment ($F_{BafA1}$). Endolysosomes coexpressing pHluorin-LAMP1 and mCherry-tagged WT or K614E OCA2 were fluorescent prior to BafA1 treatment ($F_o$), indicating that expression increased pH. Endolysosomes that coexpressed pHluorin-LAMP1 and V443I were dim at baseline ($F_o$), similar to CTL and mislocalized 5mut, representing an acidic lumen, and pHluorin-LAMP1 fluorescence increased upon BafA1 treatment ($F_{BafA1}$) (scale bar = 10 μm). (**C**) Fluorescence intensity following BafA1 treatment was compared with baseline fluorescence to determine relative baseline *Figure 3. Continued on next page*

pH regulation by OCA2 remains tentative and awaits a better understanding of melanosomal membrane conductance and signaling.

## OCA2 contributes to a melanosomal chloride conductance ($I_{melano}$) required for pigmentation

To investigate endogenous OCA2, we measured native currents by direct patch-clamp recordings from melanosomes. This is technically challenging due to the intracellular localization and small size (300–500 nm) of melanosomes and because treatment of melanocytes with vacuolin-1 did not significantly enlarge melanosomes. To circumvent these difficulties we exploited a dermal melanocyte cell line derived from mice deficient in ocular albinism 1 (*Oa1*) (*Palmisano et al., 2008*), in which melanosomes are enlarged to up to ~1 μm diameter (*Cortese et al., 2005*). Recordings from individual melanosomes dissected from $Oa1^{-/-}$ melanocytes under the same conditions used for endolysosome patch-clamp experiments (*Figure 4A*) revealed a large outwardly rectifying current ($I_{melano}$) with a negative $E_{rev}$ that was dependent on luminal Cl⁻, similar to currents recorded from OCA2-expressing endolysosomes (*Figure 4B,C*). When currents recorded in the presence of luminal Gluc⁻ were subtracted from $I_{melano}$ measured in the presence of Cl⁻, $E_{rev}$ was similar to $E_{Cl}$ (*Figure 4—figure supplement 1A*). The $E_{rev}$ for $I_{melano}$ measured with NMDG-based solutions was found to be −44.3 ± 2.5 mV for 48 mM luminal and 8 mM cytoplasmic [Cl⁻] and −2.3 ± 1.1 mV for symmetrical 48 mM [Cl⁻] (*Figure 4—figure supplement 1B,C*), consistent with the values measured for endolysosomes expressing OCA2 (*Figure 1E*) and with the Nernst potential for Cl⁻ selective currents.

In addition to skin melanocytes, melanosomes are also present in the retinal pigment epithelium of the eye. We patch-clamped melanosomes dissected from freshly isolated RPE cells from American bullfrog (*Lithobates catesbeianus*) that are larger than those from mammalian RPE, thus allowing for patch-clamp experiments (*Figure 4E*, *Figure 4—figure supplement 2*). RPE whole-melanosome recordings identified an outwardly rectifying current with a negative $E_{rev}$ that was nearly abolished (reduced by ~90%) by substituting luminal Cl⁻ with Gluc⁻ (*Figure 4E,F*). Together, these data indicate that a current with similar properties as $I_{OCA2}$ is present in melanosomes from skin and RPE.

To determine if OCA2 contributes to $I_{melano}$, we used OCA2-targeted siRNA to reduce OCA2 expression (*Figure 5—figure supplement 1A*).

*Figure 3. Continued*

acidity in enodolysosomes from CTL cells, expressing WT, K614E, V443I, OCA2 or from cells expressing mislocalized 5mut. Bars represent average normalized change in pHluorin-LAMP1 fluorescence (±s.e.m., n = 3 independent experiments, each experiment calculated as the average of n = 78–214 endolysosomes).
The following figure supplement is available for figure 3:

**Figure supplement 1**. LysoSensor pH measurements.

Consistent with previous findings (*Sviderskaya et al., 1997*), reducing endogenous OCA2 expression markedly decreased melanin content in $Oa1^{-/-}$ melanocytes (*Figure 5A*, *Figure 5—figure supplement 1B*). Melanosomes dissected from melanocytes expressing OCA2-targeted siRNA had dramatically reduced current amplitudes and melanin content compared with melanosomes from cells expressing scrambled siRNA (control), indicating that OCA2 is required for $I_{melano}$ and pigmentation (*Figure 5B,C*). Expression of WT OCA2, but not of the V443I mutant, in siRNA-treated cells was sufficient to reconstitute $I_{melano}$ and rescue melanization (*Figure 5A–C*), suggesting that the OCA2-mediated current is required for melanin production.

Might melanin itself be critical for $I_{melano}$? To address this question, we treated $Oa1^{-/-}$ melanocytes with phenylthiourea (PTU), a potent inhibitor of the key melanogenic enzyme tyrosinase, resulting in melanin depletion (*Figure 5—figure supplement 2A*). Recording from melanosomes dissected from PTU-treated melanocytes revealed a current with the same properties as in vehicle-treated cells and similar to $I_{melano}$, suggesting that melanin does not influence $I_{melano}$ (*Figure 5—figure supplement 2B,C*). We thus concluded that OCA2 expression is required for $I_{melano}$ and melanization, but melanin is not required for $I_{melano}$.

We next sought to compare the biophysical properties of the endogenous OCA2-mediated melanosome current ($I_{melano}$) in dermal and RPE melanosomes with those of the OCA2-expressing endolysosome current ($I_{OCA2}$). Excised cytoplasmic-side-out patches from partially dissected endolysosomes of OCA2-expressing AD293 cells (*Figure 6A*) exhibited single-channel currents ($i_{OCA2}$) with greatest activity at positive membrane potentials (*Figure 6B*), consistent with the outward rectification measured for whole-endolysosome $I_{OCA2}$ (*Figure 1B*). Patches from mock-transfected endolysosomes did not exhibit similar single-channel activity (*Figure 6—figure supplement 1A,B*). The $E_{rev}$ of $i_{OCA2}$ was −66 mV, close to $E_{Cl}$ predicted by the Nernst equation (−68 mV), and the unitary slope conductance was 58 ± 2 pS (*Figure 6C*).

Ion channels are pore-proteins that allow rapid diffusion of ions (>$10^6$ ions/s) down their concentration gradients, while carriers and pumps typically move ions at slower rates (<$10^5$ ions/s). We thus calculated the OCA2-mediated transport rate at 80 mV, where we measured the largest single-channel amplitude, and found a transport rate of ~5.5 × $10^7$ ions/s. The large conductance recorded in isolated patches from OCA2-expressing endolysosomes indicates that OCA2 functions as a critical component of an electrodiffusive anion channel. Similar properties were measured for single-channel currents in excised cytoplasmic-side-out patches of dermal melanosomes from $Oa1^{-/-}$ melanocytes ($i_{melano}$) ($E_{rev}$ = −67 mV and unitary slope conductance = 60 ± 1 pS) (*Figure 6D,E*). Thus, the single-channel properties of this endogenous melanosomal current are nearly identical to those recorded from OCA2-expressing endolysosomes.

To determine the anion selectivity of the recorded currents, we estimated the permeability to different anions by determining $E_{rev}$ under symmetrical concentrations (48 mM) of luminal $Cl^-$ and cytoplasmic $Cl^-$, $Br^-$, $I^-$, $F^-$ or $Gluc^-$. $E_{rev}$ for $I_{OCA2}$ was close to zero for cytoplasmic $Cl^-$ and $Br^-$ and shifted to negative voltages for $I^-$, $F^-$, and $Gluc^-$ (*Figure 6F*), suggesting that in endolysosomes OCA2 mediates a current selective for $Cl^-$ and $Br^-$, but poorly permeable to $I^-$, $F^-$, and $Gluc^-$. A similar shift in $E_{rev}$ was measured for $I_{melano}$ in dermal or RPE melanosomes (*Figure 6F*), indicating that the endogenous currents have the same selectivity profile. Moreover, the calculated permeability ratios ($P_x/P_{Cl}$) for the heterologously expressed currents were nearly the same as the endogenous ones (*Figure 6G*). Collectively, these results suggest that $I_{OCA2}$ and $I_{melano}$ are mediated by the same anion channel.

## Conclusions and implications

Our results suggest that OCA2 is an essential component of a melanosome-specific anion channel. Heterologous endolysosomal expression of OCA2 contributes to a chloride channel with biophysical properties similar to an endogenous melanosomal OCA2-mediated channel. Importantly, OCA2

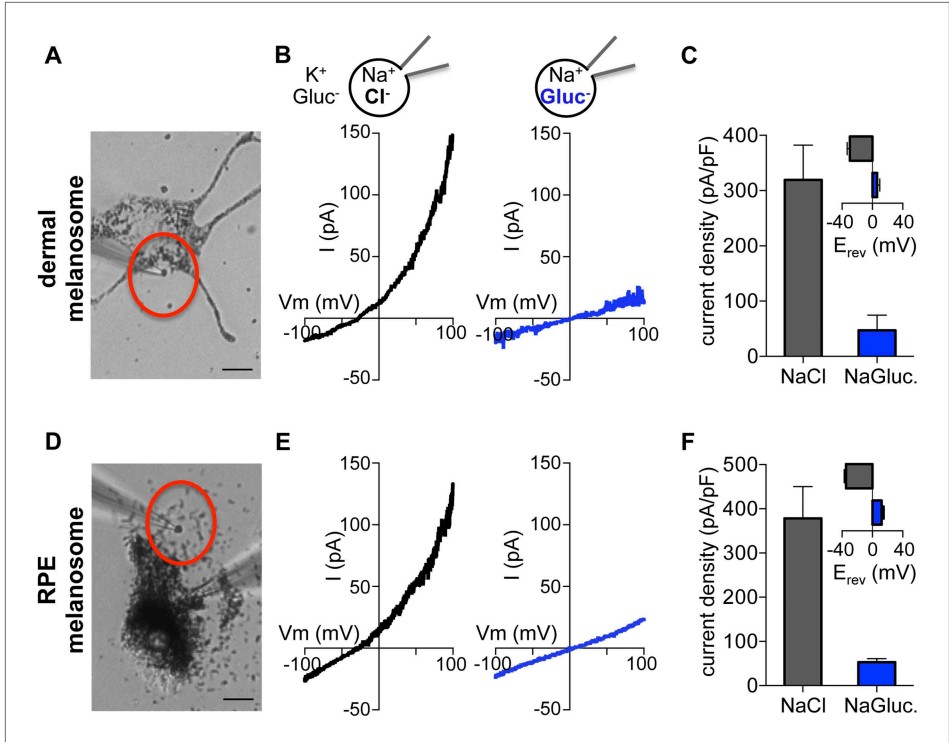

**Figure 4**. Direct recording from melanosomes identifies an endogenous Cl⁻ current ($I_{melano}$). (**A**) Dermal macromelanosomes were individually dissected from $Oa1^{-/-}$ melanocytes and patch-clamped. A NaCl-based luminal solution and KGluc-based cytoplasmic solution were used for recordings (scale bar = 10 μm). (**B**) Representative I–V relationships exhibit a Cl⁻-dependent outwardly rectifying current ($I_{melano}$) with a negative $E_{rev}$. Substituting luminal Cl⁻ for Gluc⁻ reduced the current amplitude and shifted $E_{rev}$ of $I_{melano}$. (**C**) Average current densities (pA/pF) measured at 100 mV. *Inset:* Average $E_{rev}$ (±s.e.m., n = 4–5 dermal melanosomes for each condition). (**D**) Patch-clamp experiments using freshly isolated bullfrog RPE melanosomes (scale = 10 μm). (**E**) Representative whole-RPE melanosome current–voltage (I–V) relationships in response to voltage ramps. Outwardly rectifying $I_{melano}$ was reduced and $E_{rev}$ shifted in the positive direction when luminal Cl⁻ was substituted for Gluc⁻. (**F**) Average current densities (pA/pF) measured at 100 mV. *Inset:* Average $E_{rev}$ (±s.e.m., n = 5 RPE melanosomes for each condition).

The following figure supplements are available for figure 4:

**Figure supplement 1**. $I_{melano}$ $E_{rev}$ is dependent on the Cl⁻ concentration gradient.

**Figure supplement 2**. RPE melanosome dissection.

activity controls the melanin content of melanosomes, most likely by regulating organellar pH. We propose that OCA2 contributes to a novel melanosome-specific anion current that modulates melanosomal pH for optimal tyrosinase activity required for melanogenesis.

## Materials and methods

### Cells and tissue

All cells were grown at 37°C and 5% $CO_2$, and reagents were from Invitrogen/Life Technologies (Grand Island, NY) unless stated otherwise. AD293, HeLa, and NF-SV60 fibroblasts were grown in DMEM, 10% fetal bovine serum (FBS, Atlanta Biologicals, Flowery Branch, GA) with or without 1% penicillin/streptomycin (P/S).

Immortalized mouse melanocyte cell lines melan-Ink4a (*Sviderskaya et al., 2002*) and melan-Oa1 (*Palmisano et al., 2008*) were grown in RPMI 1640, 10% FBS, 1% P/S, 200 nM phorbol 12-myristate-13-acetate (Sigma, St. Louis, MO) at 37°C and 10% $CO_2$. Primary human epidermal melanocytes and keratinocytes were isolated from neonatal foreskin. Human epidermal melanocytes (Cascade Biologics/Life Technologies) were cultured in Medium 254, Human Melanocytes Growth Supplement, and 1% P/S.

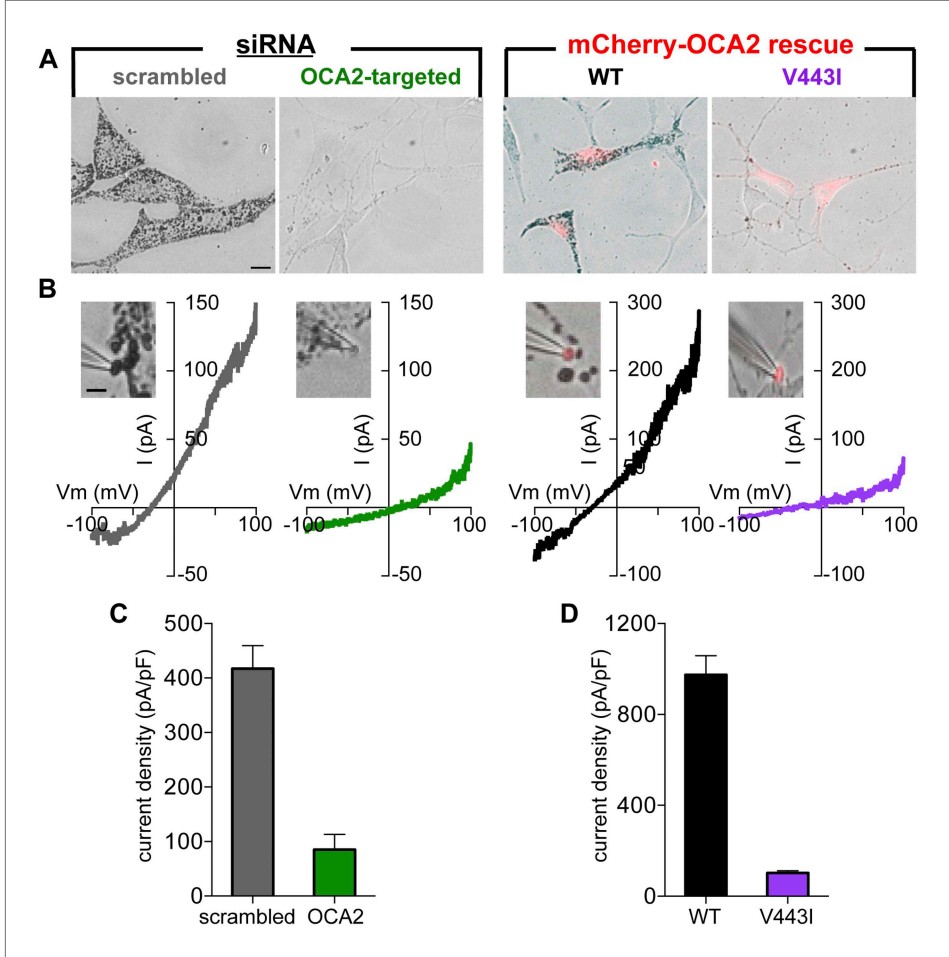

**Figure 5**. OCA2 is required for I$_{melano}$ and pigmentation. (**A**) Representative images of *Oa1$^{-/-}$* melanocytes expressing scrambled or OCA2-targeted siRNA, the latter of which significantly decreased melanin content. Pigmentation was restored by transfection with mCherry-tagged WT OCA2, but not with the mCherry-tagged V443I mutant (scale bar = 10 μm). (**B**) Representative I–V relationships from melanosomes from cells expressing scrambled siRNA, OCA2-targeted siRNA, and rescued with WT or V443I OCA2. *Insets:* images of the individual melanosomes used for the shown recordings (scale bar = 3 μm). (**C**) Average I$_{melano}$ current densities (pA/pF) measured at 100 mV in melanosomes from cells expressing scrambled siRNA or OCA2-targeted siRNA (±s.e.m., n = 4 melanosomes per condition). (**D**) Average current densities (pA/pF) measured at 100 mV in melanosomes expressing WT or V443I OCA2 from cells expressing OCA2-targeted siRNA (±s.e.m., n = 3–4 melanosomes per condition).

The following figure supplements are available for figure 5:

**Figure supplement 1**. OCA2-targeted siRNA reduces OCA2 mRNA and melanin content in *Oa1$^{-/-}$* melanocytes.

**Figure supplement 2**. Melanin is not required for I$_{melano}$.

Human epidermal keratinocytes (Lifeline Cell Technology, Frederick, MD) were cultured in Dermalife basal medium supplemented with LifeFactors and 1% P/S.

Transfection of cells used for electrophysiology and imaging was carried out using Lipofectamine 2000 (Invitrogen/Life Technologies) according to manufacturer's protocol, unless otherwise stated.

## RPE tissue

American Bullfrogs (*L. catesbeianus*) eyes were hemisected and RPE tissue was removed from the eyes with forceps after removal of the retinas. The dissection was performed in normal room light using light-adapted frogs. The tissue was kept at 4°C in a modified Ringer's solution containing (mM): 111 NaCl, 2.5 KCl, 1 CaCl$_2$, 1.5 MgCl$_2$, 0.02 EDTA, 3 HEPES, pH 7.6. Tissue was used for a maximum of 36 hr.

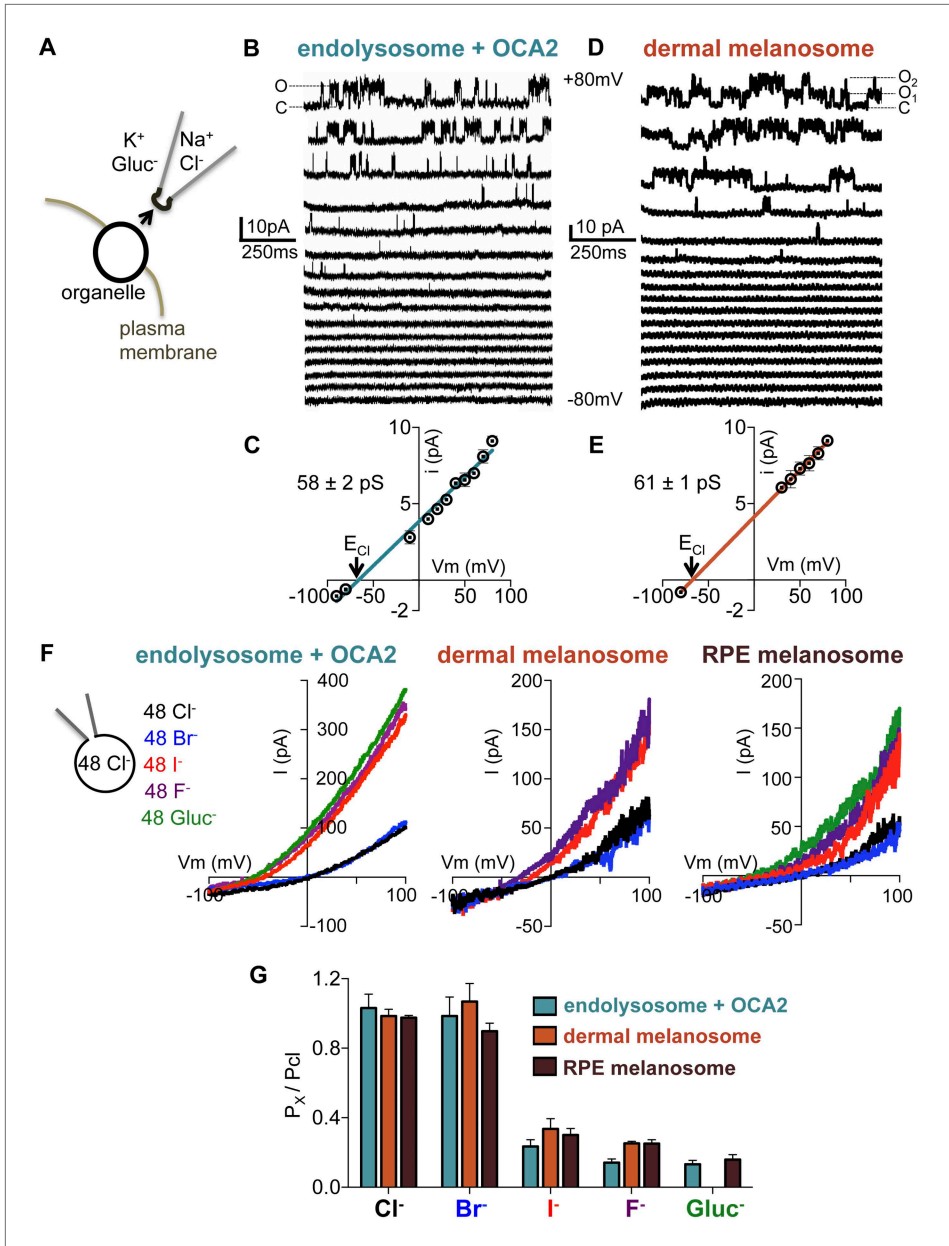

**Figure 6**. Endolyosomal $I_{OCA2}$ has the same properties as melanosomal $I_{melano}$. (**A**) Cytoplasmic-side-out patches were excised from partially dissected organelles to carry out single-channel recordings. (**B**) Representative single-channel currents recorded in response to voltage steps between −80 mV and +80 mV from a patch excised from an OCA2-expressing endolysosome (O indicates the open state and C the closed state of the channel). (**C**) Average single-channel current amplitudes from patches excised from OCA2-expressing endolysosomes (±s.e.m., n = 4 patches) ($E_{Cl}$ = Nernst potential for Cl⁻). (**D**) Single-channel currents from a representative patch excised from a dermal melanosome that contains at least two channels ($O_1$ indicates one open channel, $O_2$ indicates two channels open simultaneously, and C both channels closed). (**E**) Average single-channel current amplitudes from patches excised from dermal melanosomes (±s.e.m., n = 3 patches). (**F**) In a representative OCA2-expressing endolysosome, dermal melanosome, and RPE melanosome, currents were selective for Cl⁻ and Br⁻, but not I⁻, F⁻, or Gluc⁻. In the same organelle, currents were recorded with 48 mM luminal NMDG-Cl while substituting symmetrical concentrations of cytoplasmic anions in an NMDG-based solution. (**G**) $I_{OCA2}$ permeability ratios based on $E_{rev}$ measurements (±s.e.m., n = 3–7 organelles for each condition).

The following figure supplement is available for figure 6:

**Figure supplement 1**. OCA2-mediated single-channel current characteristics.

## Molecular biology

OCA2 tagged with GFP or mCherry was generated by inserting human OCA2 (hOCA2, NM_000275) between the BamHI/XhoI sites of pcDNA4/TO (Invitrogen/Life Technologies). K614E and V443I mutations in hOCA2 were made using site-directed mutagenesis and verified by sequencing. 5mut hOCA2 was generated by Genscript (Piscataway, NJ).

### siRNA

Mouse OCA2 (mOCA2)-targeted and scrambled siRNAs were designed and expressed using the BLOCK-iT miRNA lentiviral system (Invitrogen/Life Technologies). mOCA2 specific oligos were cloned into pcDNA6.2-GW/EmGFP-miRNA, then recombined into pLenti6/V5-DEST vector and expressed in $Oa1^{-/-}$ mouse melanocytes using viral transduction (*Bellono et al., 2013*). Stable cell lines of OCA2-targetd or control siRNA-expressing melanocytes were created using fluorescence-activated cell sorting (FACS) and subsequently kept under blasticidin selection (10 μg/ml). For rescue experiments hOCA2 was transfected in mouse melanocytes expressing mOCA2-targeted siRNA.

### Quantitative PCR

Total RNA was extracted using the RNeasy Plus kit (QIAGEN) and reverse transcribed using SuperScript III kit (Invitrogen/Life Sciences). mOCA2 mRNA levels were determined using the following primers: F 5′-CAGGAATCGCAGAGAAGCAT-3′ and R 5′-AGATAGCACATCCCAATGGTG-3′. hOCA2 mRNA levels in an array of human tissues (Origene) and in human epidermal skin cells were determined by qPCR with the following primers F 5′-GTGTGCAGGGATTGCAGAAC-3′ and R 5′-ACATCCCAACAGTGCAGGAC-3′. Reactions were prepared according to manufacturer instructions using Power SYBR Green (Applied Biosystems/Life Technologies) and cycled on a 7500 Real-Time PCR System (Applied Biosystems/Life Technologies). Actin was used as an internal control and all reactions were run in triplicate. mRNA levels were quantified by calculating $2^{-\Delta CT}$ values for each condition.

## Organellar electrophysiology

### Organelle dissection

AD293 cells were transiently transfected 1 day prior to electrophysiological recordings and treated with 1 μM vacuolin-1 (EMD Millipore, Billerica, MA) for 6–12 hr before the patch clamp experiments. Enlarged endolysosomes were individually dissected for patch clamp experiments, as previously described (*Dong et al., 2008*). In short, a borosilicate patch pipette was used to cut the cell membrane and push out individual organelles. Enlarged dermal melanosomes were dissected from $Oa1^{-/-}$ mouse melanocytes and RPE melanosomes from bullfrog RPE cells using two patch pipettes.

### Recording conditions

Organelle patch-clamp recordings were carried out at room temperature using an EPC 10 amplifier (HEKA Instruments, Lambrecht, Germany) with PatchMaster software (HEKA Instruments). Data were filtered at 2.9 kHz and digitized at 10 kHz. Membrane potentials were corrected for liquid junction potentials. Organelle currents were recorded using borosilicate glass pipettes polished to 7–8 MΩ for lysosomes and 9–12 MΩ for melanosomes. Standard pipette/lumen solutions contained (mM): 140 NaCl, 5 KCl, 1 $MgCl_2$, 2 $CaCl_2$, 10 HEPES, 10 MES, 10 glucose; pH 4.6 for endolysosomes and pH 6.8 for melanosomes, unless otherwise stated. Standard bath/cytoplasmic solution contained (mM): 140 K-gluconate, 5 NaCl, 2 $MgCl_2$, 0.39 $CaCl_2$, 1 EGTA ($Ca^{2+}$ buffered to 100 nM), 20 HEPES, pH 7.2. NMDG-based solutions were used in some experiments to block endogenous cation permeabilities. The holding potential was 0 mV and currents were measured in response to 500 ms voltage ramps from −100 to +100 mV, unless otherwise stated. Whole-organelle current density was calculated by normalizing to capacitance (0.14–0.5 pF for melanosomes, 0.24–1.2 pF for endolyosomes). Single-channel current amplitudes were measured from the middle of the noise band between closed and open states or calculated from the difference between Gaussian-fitted closed and open peaks on all-points amplitude histograms for each excised patch record. To determine if $I_{OCA2}$ passes $Cl^-$ electrodiffusively, reversal potential ($E_{rev}$) was measured using 48 mM luminal $Cl^-$ and variable cytoplasmic $[Cl^-]$ and compared with $E_{rev}$ predicted by the Nernst potential for $Cl^-$: $E_{Cl} = (RT/zF) \ln([Cl_{luminal}]/[Cl_{cytoplasmic}])$. R = gas constant, z = valence (−1 for $Cl^-$), T = absolute temperature, and F = Faraday constant.

## Anion permeability

Relative permeability of $I_{OCA2}$ was determined by measuring the shift in $E_{rev}$ after the substitution of bath/cytoplasmic anions from $Cl^-$ to $Br^-$, $I^-$, $F^-$, or $Gluc^-$. Pipette/luminal and bath/cytoplasmic solutions were 48 mM NMDG-X solutions, where X = $Cl^-$ for pipette/luminal and X = $Cl^-$, $Br^-$, $I^-$, $F^-$, or $Gluc^-$ for bath/cytoplasmic. Permeability ratios were estimated using the Goldman-Hodgkin-Katz (GHK) equation: $P_X/P_{Cl} = \exp(\Delta E_{rev}F/RT)$, where $\Delta E_{rev}$ is the difference between $E_{rev}$ in symmetrical Cl ($E_{rev}$ = 0 mV) and that in cytoplasmic X. We did not correct for the possibility of the permeation of symmetrical $NMDG^+$ or $H^+$.

## pH imaging

### pHluorin-LAMP1

AD293 cells were cotransfected with mCherry-OCA2 and pHluorin-LAMP1 1 day prior to imaging experiments. pHluorin-LAMP1 expression was verified by treatment with bafilomycin A1 (BafA1, 2 µM). pHluorin-LAMP1 fluorescence intensity was quantified in endoysosomes that coexpressed mCherry-OCA2 and pHluorin-LAMP1 by subtracting the initial fluorescence from fluorescence elicited by treatment with BafA1: $[(F_{BafA1} - F_o)/F_o]^{-1}$.

### LysoSensor DND-160

AD293 cells were transfected 1 day prior to imaging experiments with mCherry-OCA2 or LAMP1 to identify endolysosomes, and incubated with 1 µM LysoSensor-ND160 for 5 min. Lysosensor was excited at 405 nm and its emission detected at 417–483 nm (W1) and 490–530 nm (W2). The ratio of emissions (W1/W2) in endolysosomes expressing mCherry-tagged OCA2 or LAMP1 was assigned to a pH value based on a calibration curve generated prior to each experiment using solutions containing 125 mM KCl, 25 mM NaCl, 24 µM Monensin, and varying concentrations of MES to adjust the pH to 3.5, 4.5, 5, 5.5, 6.5, 7, 7.5. The fluorescence ratio was linear for pH 5.0–7.0.

## Immunofluorescence microscopy

Melan-ink4a melanocytes were seeded on coverslips coated with Matrigel (BD Biosciences, San Jose, CA) at $3–4 \times 10^4$ cells/well in a 24-well plate, transfected the next day with 0.8 µg of plasmid DNA; HeLa cells were seeded on glass coverslips at $10^5$ cells/well in a six well plate, transfected the next day using GeneJuice (EMD Millipore) as recommended by the manufacturer, with 0.1 µg of plasmid DNA. Both cell types were analyzed 48 hr post-transfection. Cells were fixed in HBSS (Invitrogen)/2% paraformaldehyde (Sigma) for 20 min at roo temperature, washed once with PBS and labeled with primary and secondary antibodies diluted in PBS with 0.2% (wt/vol) saponin, 0.1% (wt/vol) bovine serum albumin, 0.02% (wt/vol) sodium azide as described (*Calvo et al., 1999*). Nuclei were labeled with 500 ng/µl Hoescht 33342 (Sigma). Antibodies used were: mouse anti-TYRP1 (TA99/mel-5, ATCC); mouse anti-human LAMP1 (H4A3) and rat anti-mouse LAMP2 (GL2A7; both from Developmental Studies Hybridoma Bank, Iowa City, IA). Donkey antibodies specific to mouse or rat immunoglobulin and conjugated to Dylight 594 or 488 were from Jackson Immunoresearch (West Grove, PA). Cells were imaged using a 100× HCX PL APO Lens on a Leica DM IRBE microscope equipped with a Retiga Exi Fast 1394 digital camera (Qimaging, Surrey, Canada) and Improvision Openlab software (Perkin–Elmer, Waltham, MA). Sequential z-stack images separated by 0.2 µm were acquired and deconvolved using the OpenLab Volume Deconvolution module. Images from single stacks are shown. Final images were generated and insets magnified using Photoshop (Adobe, San Jose, CA). Colocalization analyses were performed using OpenLab software, as described (*Setty et al., 2007*). Briefly, images of individual cropped cells were rendered binary using the 'Density slice' module and the densely labeled perinuclear region was excluded from analysis. Pixel overlap between binary red and green images in the remaining cell regions was defined using the 'Boolean operations' module and further analyzed using Excel (Microsoft, Redmond, WA).

## Melanin quantification

Melanin from $Oa1^{-/-}$ melanocytes expressing scrambled or OCA2-targeted siRNA was quantified as previously described (*Oancea et al., 2009*). Briefly, the soluble and insoluble fractions of melanocytes were separated after cell lysis with 1% Triton X-100 (Sigma) in PBS pH 7.4. Total protein was measured in the soluble fraction using a Bradford Assay (Bio-Rad Laboratories, Waltham, MA). The insoluble fraction was dissolved in 1 N NaOH by incubation for 30 min at 80°C and used to quantify melanin by measuring the optical density of each sample at 405 nm, then fit with a standard curve generated

using synthetic melanin (Sigma). Average cellular melanin values were quantified as the ratio between total melanin and total protein from the same dish.

## Data analysis

All data are shown as mean ± s.e.m. Data were considered significant if $p < 0.05$ using unpaired two-tailed Student $t$ test or one-way ANOVA.

## Acknowledgements

We thank Drs C Cang and D Ren, who were instrumental for assisting us with the endolysosomal patch-clamp technique, V Yorgan and Dr SB Lizarraga for help with pH measurements and analyses, Dr TJ Roberts and D Sleboda for help with frogs, and Dr AL Zimmerman for advice and suggestions. We are also thankful to Dr JS Orange for providing the pHluorin-LAMP1 construct. We thank Drs AL Zimmerman, JA Kauer, and D Ren for critical reading of the manuscript. This work was supported by NIAMS (R01 AR066318 to EO and R01 AR048155 to MSM), NEI (R01 EY015625 to MSM), Brown University (EO), NIGMS training grant T32 GM077995 (NWB), a NSF Graduate Research Fellowship (NWB), and NIGMS training grant T32 GM007229 (AJL).

## Additional information

### Funding

| Funder | Author |
| --- | --- |
| National Science Foundation | Nicholas W Bellono |
| National Institute of General Medical Sciences | Ariel J Lefkovith, Nicholas W Bellono |
| National Institute of Arthritis and Musculoskeletal and Skin Diseases | Michael S Marks |
| National Eye Institute | Michael S Marks |
| Brown University | Elena Oancea |

The funders had no role in study design, data collection and interpretation, or the decision to submit the work for publication.

### Author contributions

NWB, Conception and design, Acquisition of data, Analysis and interpretation of data, Drafting or revising the article; IEE, AJL, Acquisition of data, Analysis and interpretation of data, Drafting or revising the article; MSM, EO, Conception and design, Analysis and interpretation of data, Drafting or revising the article

### Ethics

Animal experimentation: All mouse melanocytes were obtained from the Wellcome Trust Functional Genomics Cell Bank as described in the Methods. American bullfrog eyes were obtained as discarded tissue from Dr Thomas Roberts' laboratory at Brown University (protocol number 1303990009) and used in agreement with all the ethics rules and regulations.

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
