## [Decision Letter]

Thank you for sending your work entitled “An intracellular anion channel critical for pigmentation” for consideration at *eLife*. Your article has been favorably evaluated by John Kuriyan (Senior editor) and Chris Miller, who is serving as a guest editor, as well as two other reviewers, one of whom, David Clapham, has agreed to reveal his identity. Chris Miller has been in touch with you about the steps required for submission of a revised manuscript. This is a more formal letter that summarizes the decision.

The Reviewing editor and the reviewers discussed their comments before we reached this decision, and the Reviewing editor has assembled the following comments to help you prepare a revised submission.

This is a satisfying study that identifies a novel ion channel involved in regulation of acidification of melanosomes, which are lysosome-like intracellular organelles specialized for melanin pigment-producing cells. The authors start with a guess – that OCA2, an unknown-function membrane protein whose genetic disruption leads to albinism – might contribute to melanosome physiology. Accordingly, they heterologously express OCA2 in mammalian cells, showing that the protein is directed to the endolysosomes, and in exceedingly challenging patch-recording experiments document a new current appearing in dissected lysosomes. They carry out a thorough characterization of this voltage-dependent, Cl- selective current, showing that it is reduced by certain mutations associated with albinism. Moreover, they detect a similar current endogenous to melanosomes from a melanocyte cell line. The biological function of this current is addressed by experiments showing that heterologous expression of OCA2 leads to pH regulation in lysosomes similar to that known for melanosomes in melanocytes, and effect that albinism mutants fail to produce. Moreover, they show that proper pH regulation in the melanocyte line is suppressed by OCA2 siRNA.

These difficult experiments are carefully done, the analysis is rigorous, and the conclusions compelling (but see below). Moreover, the system offers novel insight into an unusual cellular physiology: pH regulation of melanin-producing organelles in melanocytes and retinal pigment epithelial cells.

The reviewers had several criticisms that will need to be addressed in a revised manuscript.

1) The conclusion that the OCA2 currents observed are mediated by an electrodiffusive ion channel mechanism is not well enough supported by the single-channel recordings of Figure 6. It is clear that these recordings are unusually difficult to make, but the absence of a statistically serious argument that these appear only upon OCA2 expression is lacking here. Since OCA2 is a member of the SLC13 family, for which only transporters are known, makes it all the more important to establish the identity of the single channels observed.

2) The use of Lysotracker to put a precise value on the lumenal pH of these lysosome-like compartments is questioned. Lysosensors are weak bases that accumulate in acidic organelles, but since this accumulation is pH dependent, as is the molecules' fluorescence it is very difficult to be confident of assessments of pH change. The calibrations are questionable because monensin alone is unlikely to allow full pH equilibration with the bathing medium.

3) The authors have given no plausible picture of why expression of a Cl- conductance should lead to alkalinization of the melanosomal lumen via the V-ATPAse, rather than acidification, and so the physiological rationale of the study needs sharpening in the Discussion (or Introduction).

4) The family of voltage-pulse-evoked currents (Figure 4) are far noisier than the ramp-evoked IV curves of Figure 4, and this discrepancy raises concerns. In addition, the currents of Figure 4 are rather redundant to those of 4E, and could be removed, but the large difference on primary data quality should be explained in the authors' response.

5) Albinism mutations. All three reported mutations presumably cause albinism, but the K614E mutation seems to generate completely normal currents. The authors never address this discrepancy, which is difficult to reconcile with the claimed function of the protein.

---

## [Author Response]

*1) The conclusion that the OCA2 currents observed are mediated by an electrodiffusive ion channel mechanism is not well enough supported by the single-channel recordings of*
Figure 6*. It is clear that these recordings are unusually difficult to make, but the absence of a statistically serious argument that these appear only upon OCA2 expression is lacking here. Since OCA2 is a member of the SLC13 family, for which only transporters are known, makes it all the more important to establish the identity of the single channels observed*.

Our argument that single channels are only present in cells transfected with OCA2 is based on a negative control (endolysosomes dissected from mock-transfected cells) of only two patches. As the reviewer indicates, this is because of the extreme difficulty of the experiments. We agree that increasing the n for the negative control would make our claim statistically stronger. In response to this, we added more recordings from cytoplasmic-side-out patches from dissected mock-transfected endolysosomes, included in a new supplemental figure (Figure 6—figure supplement 1). In this figure we show all six negative control recordings, none of which has currents similar to those measured in all eight of the OCA2-expressing cells and dermal melanosomes. We also plotted all-points amplitude histograms for each recording and graphed the average maximal amplitudes obtained from Gaussian fits of each amplitude histogram for each recording. These data are very consistent and, we believe, convincing that the single channels recorded require OCA2 expression.

*2) The use of Lysotracker to put a precise value on the lumenal pH of these lysosome-like compartments is questioned. Lysosensors are weak bases that accumulate in acidic organelles, but since this accumulation is pH dependent, as is the molecules' fluorescence it is very difficult to be confident of assessments of pH change. The calibrations are questionable because monensin alone is unlikely to allow full pH equilibration with the bathing medium*.

We agree that LysoSensor quantification is not flawless and we altered the text and figures to reflect that. However, because the LysoSensor measurements complement the pHluorin experiments, we feel that it would be useful to keep them in supplemental data (Figure 3—figure supplement 1) and adjusting the wording for the quantification of luminal pH accordingly.

*3) The authors have given no plausible picture of why expression of a Cl- conductance should lead to alkalinization of the melanosomal lumen via the V-ATPAse, rather than acidification, and so the physiological rationale of the study needs sharpening in the Discussion (or Introduction)*.

We don’t understand the mechanism by which OCA2 expression makes the endolysosomal lumen more alkaline, but can offer some of our hypotheses or speculations.

There are two determinant factors for our hypothesis/model: the organelle membrane potential (Vm) and the Nernst potential for endolysosomal/melanosomal Cl^−^ (E_Cl_).

If Vm is more positive than E_Cl_, then Cl^−^ will flow from the lumen into the cytosol. Negative charges leaving the lumen will make the inside more positive and decrease the electrogenic pumping of V-ATPase. In addition, OCA2-mediated Cl^−^ efflux will also decrease the luminal Cl^−^ concentration, decreasing the driving force for CLC antiporters, which will bring less H^+^ into the lumen. A decrease in H^+^ pumped by V-ATPases and transported by CLC would result in a more basic luminal pH.

Vm > E_Cl_ → **↑** pH lumen

If Vm is more negative than E_Cl_, then Cl^−^ will flow from the cytosol into the lumen. The Cl^−^ influx would shunt the positive-inside membrane potential, which in turn would increase the electrogenic pumping of V-ATPase, leading to H^+^ accumulation in the lumen. In this case OCA2-mediated Cl^−^ transport would acidify the lumen of the endolysosome or melanosome.

Vm < ECl → **↓** pH lumen

Our experimental evidence suggests that the first possibility applies to our data. But neither of the two values, Vm nor E_Cl_, has been unequivocally determined. In support of the assumption that Vm > E_Cl_, the lysosomal membrane potential has been estimated to be in the -30 – -10 mV range (Koivusalo et al., Traffic 2011; Steinberg et al., JCB 2010; Sonawane et al., JBC 2002, Cang et al., Cell 2013) and E_Cl_ is likely to be in the range of the typical plasma membrane value of -60mV, based on estimates of Cl^−^ concentrations in lysosomes and the cytosol (Steinberg et al., JCB 2010). Under our normal experimental conditions ([Cl^−^]_in_ = 150 mM, [Cl^−^]_out_ = 10 mM) E_Cl_ = -68 mV. In support of the second scenario (Vm < ECl), the Brilliant laboratory found the melanosomes of OCA2-deficient mice to be less acidic compared with WT using DAMP and anti-DNP staining (Puri et al., J. Invest. Dermatol. 2000). However, this change in pH cannot be directly attributed to the absence of a Cl^−^ conductance mediated by OCA2 because these melanosomes have biogenesis and maturation defects and are likely to lack other proteins important for pH and membrane voltage regulation.

Our ability to come up with a clear model is hindered by the fact that there are many unknown variables that could contribute to the regulation of organellar pH and Vm and that might be different between melanosomes and endolysosomes. The effect of the Cl^−^ conductance on the pH of the lumen could be a more complex mechanism mediated by additional channels and transporters. It is possible that melanosomes may have voltage-sensitive channels or enzymes, similar to endolysosomes (Cang et al., Nature Chem. Biol. 2014), which could be affected by OCA2 to regulate pH. In endolysosomes expressing OCA2, other ion channels might become regulated by heterologous OCA2 (TRPML, TPC, etc.) to indirectly modulate pH. In melanosomes, there are a number of membrane proteins with predicted ion transport function (SLC45A2, SLC24A5, etc.) that could contribute to this pathway by directly transporting H^+^ or by modulating H^+^ transport mechanisms. Because endolysosomes and melanosomes express different proteins that contribute to ion transport and might have different Vm, in addition to the hypothesized different luminal pH, it is also possible that OCA2 expression does not have the same effect on the pH of endolysosomes compared to the pH of melanosomes. Thus, until we better understand melanosome membrane conductance and signaling, our model depicting pH regulation by OCA2 remains highly speculative.

We now include a brief paragraph discussing our hypothetical model.

*4) The family of voltage-pulse-evoked currents (*Figure 4*) are far noisier than the ramp-evoked IV curves of*
Figure 4*, and this discrepancy raises concerns. In addition, the currents of*
Figure 4
*are rather redundant to those of 4E, and could be removed, but the large difference on primary data quality should be explained in the authors' response*.

We agree that the melanosomal recordings from Figure 4 are lower quality than the currents recorded in response to ramps. We included them with the intention of showing current kinetics in response to voltage steps. We think these recordings look so noisy because whole-melanosome recordings, in particular from RPE melanosomes, are especially unstable in response to prolonged voltage pulses, compared with ramps. We also recognize the redundancy in the data and removed Figure 4, as suggested. In place, we have added summary data from the ramp-elicited currents in RPE melanosomes (new Figure 4).

*5) Albinism mutations. All three reported mutations presumably cause albinism, but the K614E mutation seems to generate completely normal currents. The authors never address this discrepancy, which is difficult to reconcile with the claimed function of the protein*.

We were also surprised by the lack of localization or whole-organelle current phenotype for the K614E variant, and do not have a definitive answer regarding why it is associated with oculocutaneous albinism. This mutation was found in a German patient with albinism who had an additional OCA2 mutation (W679C), perhaps each mutation inherited from one parent (Passmore et al., Hum Genet 1999). A similar mutation (K614N) was found in a Tanzanian patient who also had a deletion in the second allele of the gene (Spritz et al., Am J Hum Genet 1995). In contrast, the V443I mutation is very common – it has been described in more than 10 studies (http://www.ifpcs.org/albinism/oca2mut.html) – and is, by itself, sufficient to cause oculocutaneous albinism.

Therefore, it is possible that the K614E mutation has a mild localization or functional phenotype that we are not able to detect using overexpression. Endogenous OCA2 is present at low amounts, thus even “modest” exogenous expression by viral transduction will likely increase the endogenous OCA2 levels by at least 10 fold. We speculate that the high expression levels could mask a mild defect in localization to melanosomes or in protein folding in the endoplasmic reticulum. It is also possible that the K614E mutation, found in a conserved cytoplasmic loop, affects the regulation of OCA2 function via an unknown mechanism present in the cytoplasm, which is lost during organelle dissection. But we have no way of addressing this concern in a more effective way.